# The Efficacy of Manual Therapy in Patients with Knee Osteoarthritis: A Systematic Review

**DOI:** 10.3390/medicina57070696

**Published:** 2021-07-07

**Authors:** Alexios Tsokanos, Elpiniki Livieratou, Evdokia Billis, Maria Tsekoura, Petros Tatsios, Elias Tsepis, Konstantinos Fousekis

**Affiliations:** 1Therapeutic Exercise and Sports Rehabilitation Laboratory, Physiotherapy Department, University of Patras, 25100 Egio, Greece; tsokanos.alexios@gmail.com (A.T.); niki.livieratou@gmail.com (E.L.); billis@upatras.gr (E.B.); mariatsekoura@hotmail.com (M.T.); tsepis@upatras.gr (E.T.); 2Laboratory of Advanced Physiotherapy (LAdPhys), Physiotherapy Department, University of West Attica (UNIWA), 12243 Athens, Greece; petrostatsios@hotmail.gr

**Keywords:** knee osteoarthritis, manual therapy, Mulligan technique

## Abstract

*Background and objectives*: Osteoarthritis (OA) is among the most common degenerative diseases that induce pain, stiffness and reduced functionality. Various physiotherapy techniques and methods have been used for the treatment of OA, including soft tissue techniques, therapeutic exercises, and manual techniques. The primary aim of this systemic review was to evaluate the short-and long-term efficacy of manual therapy (MT) in patients with knee OA in terms of decreasing pain and improving knee range of motion (ROM) and functionality. *Materials and Methods*: A computerised search on the PubMed, PEDro and CENTRAL databases was performed to identify controlled randomised clinical trials (RCTs) that focused on MT applications in patients with knee OA. The keywords used were ‘knee OA’, ‘knee arthritis’, ‘MT’, ‘mobilisation’, ‘ROM’ and ‘WOMAC’. *Results*: Six RCTs and randomised crossover studies met the inclusion criteria and were included in the final analysis. The available studies indicated that MT can induce a short-term reduction in pain and an increase in knee ROM and functionality in patients with knee OA. *Conclusions:* MT techniques can contribute positively to the treatment of patients with knee OA by reducing pain and increasing functionality. Further research is needed to strengthen these findings by comparing the efficacy of MT with those of other therapeutic techniques and methods, both in the short and long terms.

## 1. Introduction

Osteoarthritis (OA) is among the most common, chronic and degenerative diseases of cartilage that epidemiologically affects older people, especially women [1,2,3]. It is a common condition of the musculoskeletal system that can occur in any joint such as the upper limbs or spine, but it is mainly observed in large joints of the lower extremities, such as the hip and knee [4,5].These joints are mainly responsible for loading activities, which require smooth, successful completion and absorption of loads or vibrations [3]. In addition, gradual degeneration and loss of articular cartilage are observed during osteophyte development, inflammation of the synovial membrane and destruction of the hypochondriac bone [3,4]. The clinical characteristics of the condition include pain, stiffness, swelling, joint deformity and functional impotence, whilst at an advanced stage, muscular atrophy may also occur, which decreases patient quality of life [6].

The therapeutic techniques applied in the rehabilitation of patients with OA include therapeutic exercise, electrotherapy and manual therapy (MT). Of these techniques, MT is a hands-on physiotherapeutic approach that can decrease patients’ levels of pain and improve their functionality [7,8,9]. Manual therapists use interventions for the musculoskeletal structures of the body, such as the joints, soft tissues and nerve tissues [10,11], to stimulate several biomechanical, neurophysiological, psychological and other non-specific changes in the patient’s body that could result in positive clinical improvement of patients’ functionality [7,12].

However, the actual effects of MT on OA remain partially elucidated. This can be attributed to the different findings of the relevant research studies, which lack consensus regarding main outcomes and conclusions. Although according to several studies which investigated the effectiveness of MT in knee OA, the use of MT techniques in OA patients has been advocated, important variables such as the actual short- and long-term effects of the intervention, the dosage of the OA medication used and the effects of the combined application with other techniques (therapeutic exercise) remain unclear [13,14,15,16]. Considering the above-mentioned scientific deficit, in this systematic review, we evaluated previous studies to assess the effect of MT on knee OA symptoms. Moreover, we assessed the short-and long-term effects of MT on knee OA.

## 2. Materials and Methods

In this systematic review, we investigated the effectiveness of MT techniques in patients with knee OA. The title was formulated on the basis of the English acronym ‘PICO’ (participants, interventions, comparisons, and outcomes). Accordingly, the participants included were patients with knee OA treated with intervention using MT techniques as compared with those treated with other interventional methods to determine their effectiveness. This systematic review was conducted in accordance with the Preferred Reporting Items for Systematic Reviews and Meta-Analysis [17] and the Cochrane Handbook for Systematic Reviews of Interventions [18]. The databases used to search for scientific material were PubMed, Physiotherapy Evidence Database and Cochrane Central Register of Controlled Trials (CENTRAL). Finally, the CENTRAL database was used, including controlled studies not shown in the Medline and Embassy databases [19]. This ensured that most of the reports were covered as best as possible [20].

The above-mentioned databases were searched during the period from February 2019 to May 2019 for published English and non-English randomised controlled epidemiological studies. The keywords used were ‘knee OA’, ‘knee arthritis’, ‘MT’, ‘mobilisation’, ‘ROM’ and ‘WOMAC’. Systematic reviews and their investigations were also included in the study to reduce the possibility of omission of investigations. The eligibility criteria included randomised trials and randomised controlled trials (RCTs) involving patients of any age with clinically diagnosed OA in one knee. In the included studies [21,22,23,24,25,26], MT was applied in one of the intervention groups to determine and compare the benefits of MT in patients with knee OA. In all the studies included, an exercise programme was applied in all groups of participants to develop a more comprehensive treatment programme. This demarcation did not affect the results, as the same exercise programme was used for all the groups. Studies that had a mixed population, that is, people with and those without knee OA, and studies involving patients with diagnosed OA in both knees were excluded.

The evaluation of the studies and the avoidance of systematic errors were based on the *Cochrane Manual for Systematic Review of Interventions* [18]. The criteria for assessing the methodological quality of the surveys are presented in Table 1 and were assessed as high risk (−), low risk (+) and unspecified risk (?).

The overall quality assessment of a study resulted from the evaluation of the individual elements summarised in Table 1. Thus, the study was classified as good when all 8 criteria were marked with low risk (+) and adequate when one criterion was characterised as high-risk (−) or if two criteria were of unspecified risk (?) and, concurrently, these criteria did not affect the outcome of the results. Moreover, a study was classified as poor if two or more criteria were of high risk (−) or undying quality (?) and if the study had one high-risk criterion (−) or two unspecified risks (?). Simultaneously, these criteria likely influenced the outcome.

The outcome measures used for reporting the effectiveness of the MT techniques included the individual functional parameters of the knee joint, such as joint range of motion (ROM) and the patient’s pain sensation and overall functionality.

For the extraction of data from each study, the Cochrane Collaboration guidelines were followed. Each study was individually evaluated in terms of its qualitative and quantitative characteristics, and its data on systematic reviews, interventions and intervention outcomes were recorded. Figure 1 summarises the results of the overall search strategy (flow diagram).

## 3. Results

The search yielded 159 potentially eligible studies (Figure 1). Previous systematic reviews that studied the effectiveness of MT in patients with knee OA were also examined. Subsequently, the titles and abstracts were carefully scanned to verify their correlation with the subject of the systematic review. Articles that presented in more than one search base were removed. Thus, after the exclusion process (Figure 1), nine articles with MT as a treatment component remained. Furthermore, three of the nine studies were removed because a combined therapy was applied in the intervention group, preventing the evaluation of MT. Thus, six studies met the inclusion criteria and were included in the systematic review [21,22,23,24,25,26].

### 3.1. Characteristics of the Included Studies

All the studies that met the entry criteria and were included in the review were randomised, but only two of them included a control group, for which no therapeutic intervention was applied (Table 2).

### 3.2. Trials

In almost all the studies [21,22,23,24,26], the participants were divided into intervention groups without a control group for which no form of intervention was applied. An exception was the study of Pollard et al. [25] in which the participants were divided into intervention and control groups. Although some similarities existed, many differences most likely affected the results of the studies. The similarities between the studies included the fact that both groups (intervention and control) yielded no differences across groups at baseline, the diagnostic criteria (MRI and clinical examination) and the outcome evaluation. An exception was the research of Pollard et al. [25] that used the McGill pain questionnaire, which, having a diagnostic criterion as an outcome, cites only the visual analogue scale for pain.

However, the number of participants varied between the research of Aseer and Subramanian [22], with only 40 participants, and that of Fitzgerald et al. [23], which examined 300 people. Among the study participants, the total number of women was significantly higher than that of men, as OA has a propensity to occur in women. A significant difference was observed in the intervention duration, ranging from 2 [22,25] up to 24 weeks [26]. Finally, the frequency of re-evaluation also differed, which was performed at 12 months in some studies and 2, 4 and 6 months in others, whilst the remaining studies lacked re-evaluation.

### 3.3. Interventions

Mutlu et al. [24] evaluated the application of either mobilisation with movement (MWM) or passive joint mobilisation (PJM) or electrotherapy combined with a standard therapeutic exercise protocol in patients with knee OA. In the patients who received MT, a set of 10 repetitions of MWMs was applied to all the treatments for each patient. MWMs comprised a sustained manual glide of the tibia (medial, lateral or rotation) in various directions during active knee extension and flexion. The participants of the electrotherapy group received transcutaneous electrical nerve stimulation through four electrodes placed around the knee joint for around 20 min (set in continuous operation (110 Hz, 50 μs) and an asymmetric two-phase pulse) and underwent therapeutic ultrasonography for 5 min (frequency, 1 MHz; ultrasound power, 0.8 W/cm (power ultrasound); head-head area (sound-head area), 5 cm^2^ and irradiated area (effective radiating area), 3.5–5 cm^2^).The common therapeutic exercise programme implemented in both groups included aerobic exercise for up to 10 min, active ROM exercises from bending to extension and vice versa in 10 repetitions and 10 sets of isometric contraction strengthening exercises of the quadriceps for 10, 3 and 30 s. In addition, all the participants were given a home therapeutic exercise programme, which was recommended to be performed twice a day.

Sit et al. [26] evaluated the application of either a patellar mobilisation therapy (PMT) protocol or a conventional treatment. In patients who received PMT, a 5-min mobilisation of the patella was performed in a specific position of the knee joint once every 2 months for a total of three treatment sessions, followed by a supervised non-weight-bearing vastus medialis exercise (5 min). A home therapeutic exercise programme was also recommended twice a day with 20 repetitions/session. Only conventional treatment was used for the patients in the control group. In both groups, the same conventional treatment was applied (medication, physiotherapy, acupuncture, etc.), and the therapists or patients were allowed to provide or accept other interventions during the research.

Abbott et al. [21] compared the results between the following four groups: exercise (Ex), exercise with booster sessions (ExB), MT and exercise (MT + Ex) and MT and exercise and booster sessions (MT + Ex + B). All the participants underwent a 45-min therapeutic exercise programme that included a warm-up, an aerobic exercise for 10 min, three sets of muscle strengthening exercises for 10 repetitions for different muscle groups, passive stretching for 60 s for various muscle groups and two neuromuscular exercises for 3 min. In the patients who participated in the groups with MT, MT manipulations for 30–45 min were applied, such as knee joint mobilisation, anteroposterior and posteroanterior tibial mobilisation, patellar sliding and stretching.

Fitzgerald et al. [23] utilised the same methodological design (Ex, ExB, MT + Ex and MT + Ex + B) used by Abbott et al. [21], with slight modifications. These included 10-min warm-up aerobics and a series of strengthening, stretching and neuromuscular exercises. In addition, therapists could add strengthening or flexibility exercises for the hip or ankle joint depending on each participant’s clinical findings. The duration of the exercise session ranged from 45 to 60 min. All the patients were given a home therapy programme two or more times a week with exercises included in the supervised treatment plan and a 30-min aerobic exercise for at least three times a week. The groups that received MT were given 20-min manipulations for mobility and flexibility of the knee joints and manipulations of the soft tissues (quadriceps, rectus femoris, hind thighs, gastrocnemius and peripherals of the patella for 20 min). The core MT techniques included those specifically addressing knee joint mobility/flexibility; soft tissues of the quadriceps, rectus femoris, hamstring and gastrocnemius muscles; and peripatellar tissues. All the patients received 12 sessions. Participants with non-booster sessions received 12 sessions over 9 weeks. The participants completed 8 sessions in the first 9 weeks, two booster sessions in the fifth month, one session in the 8th month and one session in the 11th month.

Aseer and Subramanian [22] evaluated either the application of MT or a conventional program. All the participants applied the traditional programme that consisted of electrotherapy through two electrodes placed on the knee for 15 min, isometric quadriceps strengthening exercises and active mobility exercises, with instructions to perform the programme at home. In the group that received MT, gentle traction was also applied by the physiotherapist at a specific position and the knee joint of the patient continuously for 30 s for a total of four times for 2 s, with a 10-s rest period between repetitions in each session. Each participant received three sessions per week and a total of six MT sessions for 2 weeks.

Finally, Pollard et al. [25] examined the effectiveness of the Macquarie Injury Management Group (MIMG) knee protocol as compared with a control. The MIMG protocol was applied in the intervention group as a specific chiropractic protocol which was comprised of a non-invasive myofascial mobilisation procedure and an impulse thrust procedure performed on the symptomatic knee of the participants. The mobilisation procedure directed a small, sustained load and specific force to the patellofemoral articulation in a predetermined movement direction. The control group received non-force techniques and were informed that they would not feel anything. The treatment plan in each group included three treatments per week for 2 weeks.

### 3.4. Risk of Bias

The criteria for the qualitative evaluation of the studies examined the risk of systematic error. This number ranged from 5 to 6, with a maximum of 8, where a lower score means a higher risk of error. All the studies were randomised, and in four studies, the distribution sequence was concealed (Table 3). Exceptions were the research of Fitzgerald et al. [23] which showed an increased risk of error, and the study of Aseer and Subramanian [22] which showed an unspecified risk of systematic error. Although the outcome evaluators were blinded, the researchers could not be blinded. These criteria remained undefined by Pollard et al. [25]. The same study succeeded in blinding the participants, but this was unclear in the rest of the research, with three of them showing an unspecified risk of systematic error [21,22,24] and two showing a high risk of systematic error [23,26]. Two of the studies were of adequate quality [21,24], two were of unspecified quality [22,25] and two were of poor quality [23,26]. Finally, selective reporting, withdrawal and other factors were present in all the studies. Thus, the final evaluation resulted in two studies of sufficient quality [21,24], two of unspecified quality [22,25] and two of poor quality [23,26].

### 3.5. Study Findings

The research results showed a reduction in pain and an increase in functionality in the short term after application of the manual techniques (Table 4). More specifically, Kaya Mutlu et al. [24] reported the powerful benefits of MT (MWM and PJM), as opposed to electrotherapy, in reducing pain, increasing range of motion (ROM), quadriceps strength and, generally, in increasing functionality. In the long term, according to Kaya Mutlu et al. [24], combining one of the two techniques with therapeutic exercise can induce increased functionality and reduce pain, with the magnitude of the benefit ranging from small to satisfactory. In their research, Sit et al. [26] identified pain reduction andincreased ROM and functionality, which improved quality of life. They also specified that patellar mobilisation combined with therapeutic exercise significantly improved the sensation of pain and function. Fitzgerald et al. [23] analysed the effectiveness of MT without reporting detailed results with the application of manual techniques to show some therapeutic benefits in the short term. Abbott et al. [21] argued that patients’ symptoms improved when the combined therapeutic exercise and MT was applied as compared with the single therapeutic exercise. These findings indicate that pain reduces and functionality increases even in one year. Aseer and Subramanian [22] highlighted the short-term benefits of manual techniques, as they observed a significant improvement in pain sensation, satisfactory improvement in knee flexion ROM and substantial improvement in patient function and quality of life. Finally, Pollard et al. [25] highlighted the short-term therapeutic results of applying the MIMG protocol regarding pain and patient function after a two-week treatment period.

## 4. Discussion

In this systematic review, a thorough literature search and evaluation were conducted. The research characteristics were then recorded and analysed to highlight the degree of validity of the conclusions. The need to highlight MT techniques as useful treatment tools for treating knee OA was suggested on the basis of existing research results.

We observed that in all the studies evaluated in this systematic review, the researchers chose to set a therapeutic exercise programme that was common to all the participants in each study and applied it either alone or in combination with another therapeutic method. An exception was the research of Pollard et al. [25], in which the intervention group received the Macquarie Injury Management Group protocol and the control group received techniques without exerting force to induce no effect in the patient and not affect the outcome. It follows that therapeutic exercise is a tool that has been used for many years as a complementary ingredient for knee pain treatment. However, a clear conclusion about the value of the therapeutic exercise could not be deduced from the data of this systematic review because the same exercise protocol was performed by all the participants in each study without comparison with any other form of treatment.

The findings of this review collectively concur that MT applications are associated with reduced pain and increased functionality both in the short and long terms. More specifically, in their research, Kaya Mutlu et al. [24] reported the powerful benefits of MT (MWM and PJM), as opposed to electrotherapy, in reducing pain, increasing ROM, quadriceps strength and, generally, in increasing functionality. In the long term, according to the results, we can assume that combining one of the two techniques with therapeutic exercise can induce increased functionality and reduced pain, with the magnitude of the benefit ranging from small to satisfactory. Sit et al. [26] identified pain reduction and increases in ROM and functionality, which enhanced patient quality of life. They also specified that patellar mobilisation combined with therapeutic exercise significantly improved patients’ pain sensation and functionality. Fitzgerald et al. [23] supported the effectiveness of MT without reporting clear results from the application of manual techniques that showed some therapeutic benefits in the short term. Abbott et al. [21] argued that patients’ symptoms improved when the combined therapeutic exercise and MT was applied as compared with single therapeutic exercise. These findings imply that pain can be reduced and functionality can be increased even in one year. Aseer and Subramanian [22] specified the short-term benefits of manual techniques, as they found a significant improvement in pain sensation, satisfactory improvement in knee flexion ROM and considerable improvement in patient function and quality of life. Finally, Pollard et al. [25] highlighted the short-term therapeutic effects of the application of the MIMG protocol on patient pain and function after a two-week treatment period.

The positive effect of MT in patients with knee OA regarding pain reduction and ROM and functionality improvements can be predominantly attributed to the neurophysiological adaptations detected after MT, as postulated in relevant studies [10,27,28]. MT is proposed to activate a series of neurophysiological effects occurring from both the peripheral and central nervous systems. The peripheral system-mediated responses after MT include reduction in blood and serum cytokine levelsand changes in other inflammatory soup and pain-relief mediators [29,30,31]. The central nervous system responses predominantly involve supraspinal inhibitory pain mechanisms modulating pain from higher centres such as the periaqueductal grey matter, amygdala and rostral ventromedial medulla [32]. Such responses, although indirectly observed through several human and animal studies, seem to be specific to MT and do not seem to be delivered either by sham MT, control or other interventions [30,33]. Furthermore, these peripheral and central nervous system enhancements in pain reduction may, in turn, influence another cascade of more specific clinical events occurring at the joint level, such as ROM and functionality. Specifically for knee OA, animal studies support repetitive passive motion mobilisation for the improvement of joint stiffness and enhancement of the articular cartilage properties of the knee [34,35]. Thus, on the basis of the above-mentioned findings, it is not unreasonable to assume that application of MT in knee OA through initial neurophysiologically induced pain inhibitory responses may potentially enhance the mechanical properties of the affected pathoanatomic structures of the knee joint. However, further research is needed to determine the long-term effects of MT on knee OA.

Furthermore, the joint mobilisation produced through MT applications stretches the joint capsule in the sagittal plane, gently mobilises any restriction to normal movement within the limits of patient tolerance and likely loosens adhesions of the patellofemoral articulation. Manual techniques can also effectively mobilise tight myofascial thigh structures and allow greater knee mobility with less effort, restriction and pain. Moreover, the documented positive effects of applying traction on vertebral joints to reduce pain are speculated to explain the improvement of functionality when applied in the knee joint in patients with OA.

This systematic review has some limitations. One limitation is that only English language surveys were included, with the option of displaying the full text. Thus, surveys that met the other entry criteria might have been rejected from the review. In addition, a limiting factor is the relatively small number of surveys that met the inclusion criteria and the inability to find good-quality studies with a low risk (+) in all criteria for assessing methodological quality using the Cochrane Risk of Bias Tool. This is due to the nature of the intervention and the inability to blind researchers. It is important to study the long-term results of OA treatment because it concerns a large percentage of the population. The intervention duration is also a limitation, with the research mostly examining short-term therapeutic results. In this systematic review, some systematic errors were identified for each study. First, the number of therapists and place of application of the techniques can affect the results, as they prevent the generalisability of the results [24] and may cause changes in treatment quality [23]. However, correct application of techniques has been proposed to induce reliable and more valid results [22]. Moreover, OA is a chronic disease, and the need to investigate the long-term effects of therapeutic interventions is considered urgent. Therefore, a limitation in some studies is the analysis of only the short-term effects of manual techniques on this condition [22,25]. An important limiting factor is the exclusion criteria, which will determine the validity of the research. Specifically, the research of Pollard et al. [25], having loose exclusion criteria, cannot be considered as a high-quality study.

By contrast, applying strict exclusion criteria affects the research methodology [26], and the small sample in the research of Aseer and Subramanian [22] increased the probability of systematic error. A limitation in the level of research was also the sources of participants [23], where the homogeneity of the participants in terms of how they were informed affected the results. Finally, the avoidance of widely accepted questionnaires and clinical trials [25], the execution of the home exercise programme and the avoidance of analgesics [24] are factors that cannot be fully controlled by therapists, even if clear instructions and encouragement are provided for adherence to the programme.

Some of the deficits and limitations mentioned earlier are considered unavoidable. Therefore, some of the investigations were considered methodologically unsound, which may be an unfair characterisation of the studies.

## 5. Conclusions

The present review results show that MT has a positive short-term effect on the functionality of patients with knee OA. We reached this conclusion after assessment of the surveys in this review, although some of the studies were of poor or unspecified quality. Regarding the long-term benefits of MT, the research findings were inadequate for making safe and reliable conclusions owing to the study design (research duration). In conclusion, future research should focus on collecting data on long-term results by conducting qualitative research on this topic.

## Figures and Tables

**Figure 1 medicina-57-00696-f001:**
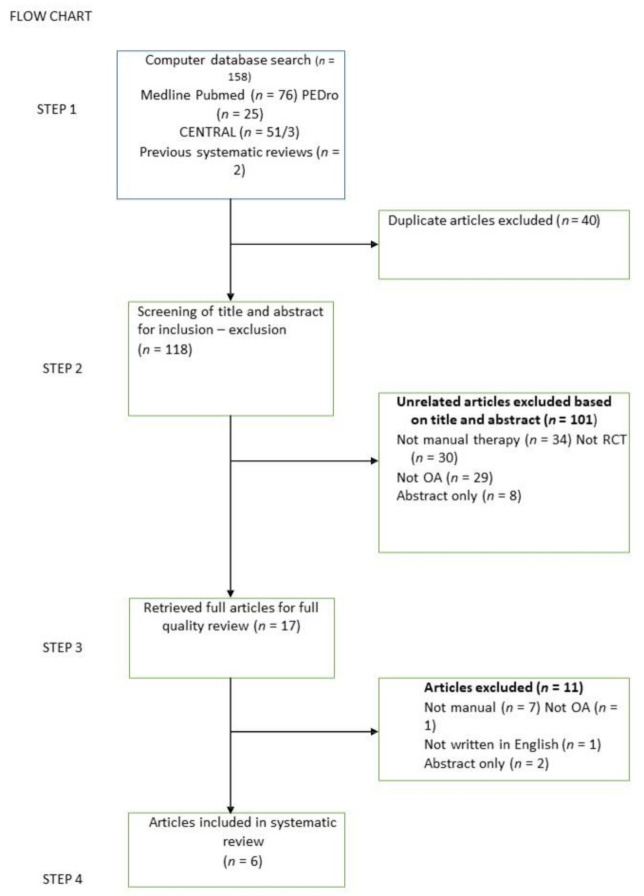
Study flowchart.

**Table 1 medicina-57-00696-t001:** Criteria for evaluating the methodological quality of the research studies [18].

Criteria for Evaluating the Methodological Quality of the Research Studies	Assessment
Random sequence generation	+/−/?
Allocation concealment	+/−/?
Blinding of the research	+/−/?
Blinding of the participants	+/−/?
Blinding of the outcome assessment	+/−/?
Selective reporting	+/−/?
Incomplete outcome data	+/−/?
Other potential threats	+/−/?

+ = low risk, − = high risk, ? = unspecified risk.

**Table 2 medicina-57-00696-t002:** Characteristics of the included studies.

Study Characteristics	Kaya Mutlu et al., 2018	Sit et al., 2018	Fitzgerald et al., 2016	Abbott et al., 2015	Aseerand Subramanian, 2014	Pollard et al., 2008
No. of participants	64 (21 MWM, 21 PJM, 22 ET)	208 (104 in the control group and 104 in the intervention group	300 (75 Ex, 76 ExB, 75 Ex+MT, 74 Ex+B+MT)	75 (Ex19, ExB 19, Ex+MT 18, Ex+B+MT 19)	40 (20 control group, 20 experimental group)	43 (26 MIMG, 17 control group)
Sex	56 females 8 males	167 females 41 males	199 females 101 males	46 females 29 males	26 females 14 males	14 females 29 males
Group distribution by sex/BMI/age	No significant differences between groups	No significant differences between groups	No significant differences between groups	No significant differences between groups	No significant differences between groups	Not reported
Dropouts	14	13	Not reported	9	0	0
Diagnostic criteria	MRI, Clinical examination	MRI and clinical examination	Not reported-	Clinical examination	MRI and clinical examination	McGill pain questionnaire
Therapeutic intervention	MWM PJM electrotherapy	PMT Conventional medication PT Acupuncture Drugs	Ex ExB MTEx MTBEx	Ex ExB Ex+MT Ex+B+MT	Pain control modalities MT Exercise	MIMG Placebo MIMG
Duration of intervention	4 weeks	24weeks	No boosters: 9 weeks Boosters: 11 months	No boosters: 9 weeks Boosters: 11 months	2 weeks	2 weeks
Outcome evaluation	WOMAC VAS ROM Muscle strength	WOMAC VAS EuroQuol-SD Clinical tests	WOMAC VAS Clinical tests	WOMAC VAS Clinical tests	VAS ROM KOOS	VAS
Reassessment	12 months	2 months–4 months–6 months	12 months	12 months	-	-

MWM = Mobilisation with movement, PJM = passive joint mobilisation, ET = electrotherapy treatment, WOMAC = Western Ontario and McMaster Universities Osteoarthritis Index, VAS = visual analogue scale, ROM = range of motion, PMT = patellar mobilisation therapy, PT = physical therapy, Ex = exercise, B = booster sessions, MT = manual therapy, MTEx = manual therapy + exercise, MTBEx = manual therapy +booster sessions + exercise, KOOS = knee injury and osteoarthritis outcome score, MIMG = Macquarie Injury Management Group.

**Table 3 medicina-57-00696-t003:** Study evaluation based on the Cochrane risk of bias tool.

	Studies					
Criteria	Kaya Mutlu et al., 2018	Sit et al., 2018	Fitzgerald et al., 2016	Abbott et al., 2015	Aseer and Subramanian, 2014	Pollard et al., 2008
Sequence generation	+	+	+	+	+	+
Allocation concealment	+	+	−	+	?	+
Blinding of the research	−	−	−	−	−	?
Blinding of the participants	?	−	−	?	?	+
Blinding of the outcome assessment	+	+	+	+	+	?
Selective reporting	+	+	+	+	+	+
Incomplete outcome data	+	+	+	+	+	+
Other potential threats	+	+	+	+	+	+
Final evaluation	Sufficient quality	Poor quality	Poor quality	Sufficient quality	Unspecified quality	Unspecified quality

+ = Yes, − = No, ? = Not reported.

**Table 4 medicina-57-00696-t004:** Study findings.

Results	Kaya Mutlu et al., 2018	Sit et al., 2018	Fitzgerald et al., 2016	Abbott et al., 2015	Aseer and Subramanian, 2014	Pollard et al., 2008
Control group	-	No Significant improvement	-	-	No significant improvement	No significant improvement
Intervention group	-	Significant improvement at: WOMAC, clinical tests	-	-	Significant improvement of pain	Significant improvement of pain, functionality and knee joint stiffness
PJM	Significant improvement of ROM, the quadriceps, strength and pain	-	-	-	-	-
MWM	Significant improvement of ROM, the quadriceps, strength and pain	-	-	-	-	-
Electrotherapy group	No significant improvement of ROM, the quadriceps, strength and pain	-	-	-	-	-
Ex	-	-	No significant improvement	No significant improvement	-	-
ExB	-	-	No significant improvement	Significant improvement compared with Ex	-	-
MT + Ex	-	-	Significant improvement in WOMAC	Significant improvement compared with Ex and the four groups Better results in terms of pain	-	-
MT + Ex + B	-	-	Significant improvement in WOMAC	No significant improvement	-	-

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
