# Peer review of "The Efficacy of Manual Therapy in Patients with Knee Osteoarthritis: A Systematic Review"

_medicina, 2021, doi:10.3390/medicina57070696_

Round 1

Reviewer 1 Report

The article reviews the literature and describes the impact of manual therapy on the efficacy of OA therapy. The authors describe an important topic because people with OA constitute a large population, and conservative treatment in these patients is essential. The work was written interestingly. The authors are aware of certain limitations. In my opinion, the manuscript requires editing. Sometimes two words are combined into one, as in the examples below. So the text needs to be edited corectly.

Line 58: theintervention

Line 71: possibilityof

Line 86: studyresulted

Line 95: includedknee

Table 2: row „sex”

Line 144: orPassive

Line 181: exercisesession

Lines 217-218: remainedundefined

Author Response

Suggestions for Authors

The article reviews the literature and describes the impact of manual therapy on the efficacy of OA therapy. The authors describe an important topic because people with OA constitute a large population, and conservative treatment in these patients is essential. The work was written interestingly. The authors are aware of certain limitations. In my opinion, the manuscript requires editing. Sometimes two words are combined into one, as in the examples below. So the text needs to be edited corectly.

Authors response: We should thank the referee for his kind words. He is absolutely right that the text needed some corrections in terms of grammar and flow.

The text was corrected based on the referee's instructions (sent for 2nd time to update and improve its grammar, formatting and flow in an academic editing company – SCRIBENDI EDITING SERVICES)

Reviewer 2 Report

Thank you to the authors for the systematic review they performed on this topic.  From my review, it does not appear as though the literature search is complete.  The choice of search terms appears to be incomplete as only abbreviated terms are used and not full words (e.g. MT instead of "Manual Therapy") also it is not clear why "knee arthritis" is searched instead of "knee osteoarthritis.  Full word search terms should be added to make sure that no articles have been missed. From what is provided most of the studies that end up in the systematic review are small in size and only 2 of 6 are considered of "sufficient" quality.   Also it not clear why unilateral OA is evaluated and not unilateral and bilateral OA. This needs to be explained better in the study selection process. Also there is no information given on baseline disease severity of the population relative to WOMAC scores and/or K-L grading which is likely to have considerable impact on the interventions being studied and their effectiveness.  If available, this should be included in the baseline characteristics and if not available explained in the discussion, the weakness this adds to the review. The literature search should be more comprehensive and avoid terms like "significantly higher" as there are no adjustments for multiplicity in comparing results across studies.  Proofing for spelling, formatting and grammar errors should also be done before resubmitting.

Author Response

We should thank the referee for his kind words and for his corrections-suggestions. He is also absolutely right that the text needed some corrections in terms of grammar and speech flow.

Response to the referres point

Referee point 1.From my review, it does not appear as though the literature search is complete.  The choice of search terms appears to be incomplete as only abbreviated terms are used and not full words (e.g. MT instead of "Manual Therapy") also it is not clear why "knee arthritis" is searched instead of "knee osteoarthritis.  Full word search terms should be added to make sure that no articles have been missed.

Authors response: The keywords used for OA were both knee arthritis and knee osteoarthritis. The term knee arthritis was used in conjunction with knee osteoarthritis to come up with a volume of research related to the object we were looking for. This led to a more comprehensive range of research studies on the subject matter.

Referee point 2.From what is provided most of the studies that end up in the systematic review are small in size and only 2 of 6 are considered of "sufficient" quality.   Also it not clear why unilateral OA is evaluated and not unilateral and bilateral OA. This needs to be explained better in the study selection process.

Authors response: Indeed, the size and quality of the surveys are not considered sufficient. This, however, is meticulously analyzed and reported in the review in several parts (limitations). The unilateral OA was preferred over the bilateral OA to ease the procedure and better control its limitations (control and comparison of the two lower extremities).

Referee point 3. Also there is no information given on baseline disease severity of the population relative to WOMAC scores and/or K-L grading which is likely to have considerable impact on the interventions being studied and their effectiveness.  If available, this should be included in the baseline characteristics and if not available explained in the discussion, the weakness this adds to the review.

Authors response: Although good tools for categorizing the severity of OA, Womac K-L grading have not been adopted in all the surveys reviewed and can not be included in the review. We chose to present variables that exist in the vast majority of the surveys we evaluated so that our results are as valid as possible.

Referee point 4.The literature search should be more comprehensive and avoid terms like "significantly higher" as there are no adjustments for multiplicity in comparing results across studies. 

Authors response: The referee is right about using the term significantly "significantly higher", which directly does not give any advantage in searching for OA studies. On the other hand, it does not have a disadvantage as it does not significantly affect the search and the final selection of the surveys. In our opinion, all the actions were done so that the research search is valid and lead to valid results.

Referee point 5. Proofing for spelling, formatting and grammar errors should also be done before resubmitting.

Authors response: The text was corrected based on the referee's instructions (sent for 2nd time to correct and improve its grammar, formatting and flow in an academic editing company – SCRIBENDI EDITING SERVICES)

Round 2

Reviewer 2 Report

Thank  you to the authors for responding to my comments and review and improving the readability of the manuscript. All comments have been addressed.